# Frequency Invariant Beamforming for a Small-Sized Bi-Cone Acoustic Vector–Sensor Array

**DOI:** 10.3390/s20030661

**Published:** 2020-01-24

**Authors:** Erzheng Fang, Chenyang Gui, Desen Yang, Zhongrui Zhu

**Affiliations:** 1Acoustic Science and Technology Laboratory, Harbin Engineering University, Harbin 150001, China; fangerzheng@hrbeu.edu.cn (E.F.); acousticmen_hrbeu@163.com (D.Y.); zhuzhongrui@hrbeu.edu.cn (Z.Z.); 2College of Underwater Acoustic Engineering, Harbin Engineering University, Harbin 150001, China; 3Key Laboratory of Marine Information Acquisition and Security (Harbin Engineering University), Ministry of Industry and Information Technology, Harbin 150001, China

**Keywords:** bi-cone acoustic vector-sensor array (BCAVSA), the mechanical coupling of *Ormia ochracea*’s two ears, frequency invariant beamforming, nested cylindrical acoustic vector-sensor array

## Abstract

In this work, we design a small-sized bi-cone acoustic vector-sensor array (BCAVSA) and propose a frequency invariant beamforming method for the BCAVSA, inspired by the *Ormia ochracea*’s coupling ears and harmonic nesting. First, we design a BCAVSA using several sets of cylindrical acoustic vector-sensor arrays (AVSAs), which are used as a guide to construct the constant beamwidth beamformer. Due to the mechanical coupling system of the *Ormia ochracea*’s two ears, the phase and amplitude differences of acoustic signals at the bilateral tympanal membranes are magnified. To obtain a virtual BCAVSA with larger interelement distances, we then extend the coupling magnified system into the BCAVSA by deriving the expression of the coupling magnified matrix for the BCAVSA and providing the selecting method of coupled parameters for fitting the underwater signal frequency. Finally, the frequency invariant beamforming method is developed to acquire the constant beamwidth pattern in the three-dimensional plane by deriving several sets of the frequency weighted coefficients for the different cylindrical AVSAs. Simulation results show that this method achieves a narrower mainlobe width compared to the original BCAVSA. This method has lower sidelobes and a narrower mainlobe width compared to the coupling magnified bi-cone pressure sensor array.

## 1. Introduction

Interest in the measurement of the radiated noise has increased considerably in the last two decades, driven by a recognition of safeguarding the national marine safety. The early method used a single pressure sensor to measure the radiated noise [1]. However, as the development of vibration reduction and noise reduction technology, the radiated noise level has decreased significantly. The early method cannot extract the information of the radiated noise as it cannot provide the spatial gain. In order to obtain a higher gain, an array composed of many pressure sensors is usually used in the sonar system, which leads to a high cost. Acoustic vector sensors (AVSs) can simultaneously measure the acoustic pressure as well as two or three orthogonal particle velocity components at a single spatial point [2,3,4,5,6,7,8,9]. As the AVS can make use of the extra available acoustic particle velocity information, the arrays composed of AVSs have some attractive advantages compared to the pressure sensor arrays with the same configuration, such as higher spatial gain [10], stronger ability to suppress noise [11], lower direction-of-arrival (DOA) estimation error [3,12], etc. Owing to the above advantages, the measurement of the radiated noise by using the acoustic vector-sensor array (AVSA) has attracted broad attention.

The radiated noise produced by underwater targets usually consists of broadband and narrowband components. As the broadband signals carry more information of the underwater targets compared to the narrowband signals, the research concern has turned to the broadband signals. Beamforming is the most used spatial processing technique among the array signal processing techniques. However, the beamwidth of the traditional beamforming becomes narrower as the signal frequency increases. When the AVSA receives the broadband signal radiated from the volume element, except for the spindle direction of the mainlobe, the incident direction locating in the other directions will lead to the signal spectra distortion [13]. In order to remedy this shortcoming, the frequency invariant beamformers which could guarantee the width of the mainlobe holding invariant with the changing of the signal frequency were developed for maintaining the signal spectra non-distortion as long as the signal is incident to the AVSA from the directions of the mainlobe. There are several techniques for designing the constant beamwidth beamformer over the desired broadband. In [14], the theory and design methods of forming a frequency invariant far-field beampattern were proposed by using the relationship of a continuously distributed sensor array’s aperture and frequency, which can be used for the one-, two-, and three-dimensional arrays. Based on the three- and four-dimensional Fourier transformation of their spatial and temporal parameters, the authors in Reference [15] proposed the design method of the frequency invariant beampattern for the two- and three-dimensional arrays that are composed of the linear sensor array. In [16], the method of selecting a reference beam is proposed for the fast Fourier-transformation-based frequency invariant beamforming, which is determined by the number of sensors and the ratio of the highest frequency and lowest frequency of the signal. However, these methods are concentrated at the linear arrays or the two- and three-dimensional array composed of linear arrays, where the disadvantage is that, when the direction of the mainlobe changes from the broadside to the endfire, the resolution becomes lower as the width of the mainlobe turns wider. Huang et al. used the Jacobi Anger expansion to approximate the beampattern of the uniform circular array and then designed the symmetric directivity pattern with the frequency invariant property, where the beamwidth of the mainlobe was invariant at any look direction in the sensor plane [17]. Although the beampattern function in [17] is independent from the signal frequency theoretically, it is subjected to the number of the sensors and the radius. Therefore, this method in [17] only exhibits a bandpass characteristic [18]. In order to obtain a frequency invariant beampattern over a large bandwidth, the authors in [19,20] used a class of uniform concentric circular arrays to design the frequency invariant beamforming and utilized the second-order cone programming to design the compensation filters, which led to a high running time of solving compensation filters due to the usage of the optimal toolbox. In [21,22,23], the harmonic nesting method was used to design the constant beamwidth pattern, where the principle was to design the frequency weighted coefficients to compensate the two subarrays with appropriate apertures in order to form a constant beamwidth over an octave. The researchers in References [24,25] extended the harmonic nesting method to linear AVSAs to obtain the frequency invariant beampattern over the octaves, which was superior to linear pressure sensor arrays in terms of spatial gain and anti-starboard ambiguity. Though the main disadvantage of the harmonic nesting method is the complex structure of the sensor arrays, the physical meaning is clear, and it can control the constant beamwidth beampattern perfectly.

It is worth noting that the above-mentioned methods focus on the planar arrays, the volumetric array composed of linear arrays, or the linear AVSAs. Currently, there are few studies on the volumetric AVSA. On the other hand, the above-mentioned methods of designing frequency invariant beamformers work well under appropriate intersensor spacing. However, the frequency band of the radiated noise from underwater targets generally falls within dozens to one thousand Hertz. This indicates it is a requirement for the sensor array with half of the wavelength interelement spacing to guarantee the desired beamwidth of the beamforming, resulting in a large aperture sensor array. In practice, large aperture sensor arrays are not usually used due to the disadvantages of high cost, difficult deployment, a high requirement for the application environment, and the ancillary facilities. Compared to large aperture sensor arrays, the small-sized sensor arrays are more suitable for applying to sonar systems. Thus, this paper aims to design a frequency invariant beamformer with a narrow beamwidth and low sidelobes for a small-sized volumetric AVSA.

The hearing system of the parasitoid fly *Ormia ochracea* gives us a hint to get out of this dilemma. To perpetuate the species, the female *Ormia ochracea* has to find a male cricket by using the cricket host’s call. The *Ormia ochracea* features an advantage of locating these crickets accurately by using its two-ear differences in intensity and arrival time from an incident acoustic wave. This phenomenon was surprising to the researchers as the distance between the *Ormia ochracea*’s two ears is only about 1.2 mm, and the wavelength of the cricket’s call is about 7 cm, which is a significant mismatch to the distance of the *Ormia ochracea*’s ears [26]. Studies in [27,28] explained that the reason about the *Ormia ochracea* creating this advantage was that the *Ormia ochracea*’s left and right acoustical-sensory organs were not physically separated, but connected by a mechanical coupling, which increased the two-ear time delay and amplitude difference. Recently, the mechanical system of the *Ormia ochracea*’s two ears has drawn sufficient attention and has been widely applied in the manufacture of microphones [29,30,31], the design of two-element antenna arrays [32,33], the DOA estimation based on a microphone array or an antenna array [34,35], etc. To our knowledge, this mechanical system of the *Ormia ochracea*’s two ears has not been studied for applications of AVSAs.

In this paper, we provide a frequency invariant beamforming method for the small-sized bi-cone AVSA (BCAVSA), which is inspired by the coupling magnification mechanism of the *Ormia ocharcea* and the harmonic nesting method. First, we design the BCAVSA using several groups of cylindrical AVSAs with different radii and heights, where the cylindrical AVSA is composed of two uniform circular AVSAs (UCAVSAs), which is used to guide the construction of the constant beamwidth beamformer. Specifically, we derive the coupling magnified matrix for the cylindrical AVSAs in order to increase the distance of the two adjacent AVSs located in the same UCAVSA, as well as that of the two UCAVSAs. Therefore, a virtual BCAVSA is obtained. Then, the idea of the harmonic nesting is extended to the BCAVSA, and the frequency invariant beamforming method for the virtual BCAVSA is developed. This method can form the frequency invariant beampattern in the three-dimensional space, and the beamwidth in the azimuth plane is invariant when the elevation angle is fixed. Owing to the coupling magnified system, the beamwidth of the beampattern based on the proposed method is narrower than that of using the original BCAVSA. Additionally, compared to the bi-cone pressure sensor array, the proposed method has lower sidelobes and a narrower width of the mainlobe.

The remainder of this paper is organized as follows: In Section 2, the background is introduced, which includes the measurement model of the small-sized BCAVSA and the response of the mechanical coupling system of the *Ormia ocharea*. In Section 3, the proposed method is derived. In Section 4, the numerical simulations are presented. Finally, we provide conclusions in Section 5.

## 2. Background

### 2.1. Measurement Model of BCAVSA

Incipiently, the wireless area proposed the bi-cone antenna array [36,37,38], which features the higher radiation power, flexible radiation pattern, etc. Later, the U.S. Navy developed a bi-cone sensor array measurement system to measure the radiated noise [39]. In the underwater, the broadband of the far-field radiated noise usually falls within dozens to a thousand of Hertz. In order to efficiently measure the low-frequency radiated noise, the interelement spacing of the bi-cone sensor array should be fitted to 0.5λ (λ is the signal wavelength). However, the size of the bi-cone sensor array is always restricted to some practical application platform such as the shallow sea. In this condition, the element spacing of the bi-cone sensor array is smaller or much smaller than the 0.5λ. On the other hand, the beamwidth of beamforming is wider with the signal frequency decreasing, which can lead to the signal spectra distortion [13]. Thus, this paper aims to propose a method for a small-sized bi-cone sensor array to form the constant beamwidth pattern to measure the low-frequency radiated noise.

Inspired by the bi-cone sensor array in [39], a small-sized BCAVSA is designed for forming the constant beamwidth pattern. It is noted that, in this paper, the "small-sized BCAVSA" means the interelement spacing is much smaller than the half of wavelength. As shown in Figure 1, the BCAVSA is composed of *Q* groups of cylindrical AVSAs. The *q*th cylindrical AVSA with the radius $r_{q}$ and height $h_{q}$ consists of the *q*th and −qth UCAVSAs, where $h_{q}=2z_{q}$, $z_{q}$ is the position of the center of the *q*th UCAVSA on the *z*-axis, 1≤q≤Q. The *q*th UCAVSA is constituted of *N* AVSs, where the *m*th AVS is located at $\left\{r_{q}\cos\psi_{m},r_{q}\sin\psi_{m},z_{q}\right\}$, where $\psi_{m}$ is the angular position measured counterclockwise from the *x*-axis, 1≤m≤N, 1≤q≤Q. Directions of three acoustic particle velocity components of AVSs are oriented along the *x*-, *y*-, and *z*-axes, respectively. It is assumed that a planar wave impinges on the BCAVSA from $\left(\theta_{s},\phi_{s}\right)$, where $\theta_{s}\in$ [0°, 360°] and $\phi_{s}\in$ [0°, 180°] denote the azimuth and elevation angles, respectively. Taking the origin of the coordinate system as the reference point, the steering vector of the *q*th UCAVSA at the signal frequency *f* can be expressed as (q=1,2,⋯,Q):
(1)$$a_{v,q}\left(f,\theta_{s},\phi_{s}\right)=a_{p,q}\left(f,\theta_{s},\phi_{s}\right)⊗u\left(\theta_{s},\phi_{s}\right),$$
where $a_{p,q}\left(f,\theta_{s},\phi_{s}\right)=a_{pc,q}\left(f,\theta_{s},\phi_{s}\right)e^{jkz_{q}\cos\phi_{s}}$ is the steering vector related to the acoustic pressure components belonging to the *q*th UCAVSA, $a_{pc,q}\left(f,\theta_{s},\phi_{s}\right)={\left[a_{pc,q,1}\left(f,\theta_{s},\phi_{s}\right),\cdots ,a_{pc,q,M}\left(f,\theta_{s},\phi_{s}\right)\right]}^{T}$, $a_{pc,q,m}\left(f,\theta_{s},\phi_{s}\right)=e^{jkr_{q}\sin\phi_{s}\cos\left(\psi_{m}-\theta_{s}\right)}$, k=2πf/c is the wave number, *c* is the sound speed in the underwater, ${\left(\;\cdot \;\right)}^{T}$ denotes the transpose operation, and $j=\sqrt{-1}$ is the imaginary unit. ⊗ denotes the Kronecker product. $u(\theta_{s},\phi_{s})$ is the 4×1 the array manifold vector of the single AVS that is independent from the signal frequency and can be expressed as:(2)$$u\left(\theta_{s},\phi_{s}\right)={\left[1,\cos\theta_{s}\sin\phi_{s},\sin\theta_{s}\sin\phi_{s},\cos\phi_{s}\right]}^{T}.$$

According to Equation (Equation 1), the steering vector of the *q*th cylindrical AVSA composed of the −qth and *q*th UCAVSA can be written as: (q=1,2,⋯,Q):(3)$$\begin{array}{c} a_{v,\mp q}\left(f,\theta_{s},\phi_{s}\right)={\left[a_{v,-q}\left(f,\theta_{s},\phi_{s}\right),a_{v,q}\left(f,\theta_{s},\phi_{s}\right)\right]}^{T}\\ ={\left[a_{p,-q}\left(f,\theta_{s},\phi_{s}\right)⊗u\left(\theta_{s},\phi_{s}\right),a_{p,q}\left(f,\theta_{s},\phi_{s}\right)⊗u\left(\theta_{s},\phi_{s}\right)\right]}^{T}\\ =a_{z,q}\left(f,\phi_{s}\right)⊗a_{pc,q}\left(f,\theta_{s},\phi_{s}\right)⊗u\left(\theta_{s},\phi_{s}\right), \end{array}$$
where $a_{p,-q}\left(f,\theta_{s},\phi_{s}\right)=a_{pc,q}\left(f,\theta_{s},\phi_{s}\right)e^{-jkz_{q}\cos\phi_{s}}$ and $a_{z,q}\left(f,\phi_{s}\right)={\left[e^{-jkz_{q}\cos\phi_{s}},e^{jkz_{q}\cos\phi_{s}}\right]}^{T}$.

At the signal frequency *f*, the steering vector of the BCAVSA can be expressed as:(4)$$a_{v}\left(f,\theta_{s},\phi_{s}\right)={\left[a_{v,\mp 1}\left(f,\theta_{s},\phi_{s}\right),\cdots ,a_{v,\mp Q}\left(f,\theta_{s},\phi_{s}\right)\right]}^{T}.$$

It can be seen from Equations (Equation 3) and (Equation 4) that the steering vector of the BCAVSA is composed of the *Q* steering vectors associated with *Q* cylindrical AVSAs, and the steering vector of each cylindrical AVSA is the Kronecker product of steering vectors those correspond to a two-element pressure sensor array along the *z*-axis, a uniform circular pressure sensor array, and a single AVS.

Note that in practice the closely spaced AVSs in passive sonar systems can encounter the undesired electromagnetic coupling. In this work, we assume that the calibration for the closely spaced AVSs is accomplished beforehand, viz., the performance of the passive sonar system will not be affected, and hence we ignore the effect of the undesired electromagnetic coupling in this work.

### 2.2. Mechanical Coupling System of Ormia ochracea’s Ears

The female *Ormia ochracea*’s hearing organ is so small that its two-ear distance is only about 1.2 mm, but, magically, it can locate crickets quite accurately by using the cricket hosting’s calling (wavelength is about 7 cm). References [27,28] show that the *Ormia ochracea*’s bilateral tympanal membranes are not physically separated but connected by a mechanical coupling which increases two-ears time delay and amplitude difference. This is the main reason that the *Ormia ochracea* can use their ears to find crickets. The mechanical coupling model of the *Ormia ochracea*’s hearing organ is given in [28], as shown in Figure 2.

In this system, the spring $\alpha_{i}$ and the dash-pot $\beta_{i}$ (i=1 or 2) approximately represent the dynamical properties of the left or right tympanal membrane, sensory organs, and surrounding structures in the *Ormia ochracea*’s two ears, the effective mass $m_{0}$ denotes all moving parts on the left/right unilateral of the intertympanal bridge, and the intertympanal bridge is composed of two rigid bars connected at the pivot by a coupling spring $\alpha_{3}$ and the dash-pot $\beta_{3}$. When a plane wave from the cricket host’ call (signal frequency is about 5000 Hz) is incident to the eardrums, the signals at the bilateral tympanal membranes $x_{1}\left(t\right)$ and $x_{2}\left(t\right)$ can be treated as the two input signals of the mechanical coupling system. Let $y_{1}\left(t\right)$ and $y_{2}\left(t\right)$ denote the moving displacements of the tympanal membranes, which are also the output of the mechanical coupling system. It is noted that this mechanical coupling system can be seen as a two-input two-output filter system. Except for the effect of $x_{1}\left(t\right)$ and $x_{2}\left(t\right)$, when the tympanal membranes move, tympanal membranes are affected by the inertia, damping, and recovery forces. According to Newton’s second law of the motion, the dynamic equation of this mechanical structure can be built as:(5)$$\begin{array}{c} \left[\begin{array}{cc} \alpha_{1}+\alpha_{3} & \alpha_{3}\\ \alpha_{3} & \alpha_{2}+\alpha_{3} \end{array}\right]\left[\begin{array}{c} y_{1}\left(t\right)\\ y_{2}\left(t\right) \end{array}\right]+\left[\begin{array}{cc} \beta_{1}+\beta_{3} & \beta_{3}\\ \beta_{3} & \beta_{2}+\beta_{3} \end{array}\right]\\ \left[\begin{array}{c} {\dot{y}}_{1}\left(t\right)\\ {\dot{y}}_{2}\left(t\right) \end{array}\right]+\left[\begin{array}{cc} m_{0} & 0\\ 0 & m_{0} \end{array}\right]\left[\begin{array}{c} {\ddot{y}}_{1}\left(t\right)\\ {\ddot{y}}_{2}\left(t\right) \end{array}\right]=\left[\begin{array}{c} x_{1}\left(t\right)\\ x_{2}\left(t\right) \end{array}\right], \end{array}$$
where ${\dot{y}}_{i}\left(t\right)$ and ${\ddot{y}}_{i}\left(t\right)$ (i=1,2) denote the speed and the acceleration of the dynamic displacements of bilateral tympanal membranes. The three terms in the left of Equation (Equation 5) denote the recovery, damping, and inertia forces, respectively.

In order to solve Equation (Equation 5) and obtain the transfer function of the mechanical coupling system, the Fourier transform is used to Equation (Equation 5) with the assumption of zero initial values, and the following can be obtained:(6)$$\begin{array}{c} \left[\begin{array}{c} Y_{1}\left(f\right)\\ Y_{2}\left(f\right) \end{array}\right]=\frac{1}{G(f)}\left[\begin{array}{cc} D_{1}\left(f\right) & -D_{3}\left(f\right)\\ -D_{3}\left(f\right) & D_{2}\left(f\right) \end{array}\right]\left[\begin{array}{c} X_{1}\left(f\right)\\ X_{2}\left(f\right) \end{array}\right], \end{array}$$
where $X_{i}\left(f\right)$ and $Y_{i}\left(f\right)$ are the Fourier transformation of the $x_{i}\left(t\right)$ and $y_{i}\left(t\right)$ with i=1,2, respectively. According to Ref. [35], $D_{1}\left(f\right)=\alpha_{1}+\alpha_{3}+j2\pi f\left(\beta_{3}+\beta_{1}\right)-m_{0}{\left(2\pi f\right)}^{2}$, $D_{2}\left(f\right)=\alpha_{2}+\alpha_{3}+j2\pi f\left(\beta_{3}+\beta_{2}\right)-m_{0}{\left(2\pi f\right)}^{2}$, and the coupling effect $D_{3}\left(f\right)=j2\pi f\beta_{3}+\alpha_{3}$. $G\left(f\right)=D_{1}\left(f\right)D_{2}\left(f\right)-D_{3}^{2}\left(f\right)$.

Let $Z_{1}\left(f\right)=X_{1}\left(f\right)-X_{2}\left(f\right)$ and $Z_{2}=X_{1}\left(f\right)+X_{2}\left(f\right)$ denote the difference and the sum of the two acoustic pressure signals. Then, Equation (Equation 6) can be further written as:(7)$$\begin{array}{c} \left[\begin{array}{c} Y_{1}\left(f\right)\\ Y_{2}\left(f\right) \end{array}\right]=\frac{1}{2G(f)}\left[\begin{array}{cc} B_{1}\left(f\right) & B_{2}\left(f\right)\\ B_{3}\left(f\right) & B_{4}\left(f\right) \end{array}\right]\left[\begin{array}{c} Z_{1}\left(f\right)\\ Z_{2}\left(f\right) \end{array}\right], \end{array}$$
where $B_{1}\left(f\right)=D_{1}\left(f\right)+D_{3}\left(f\right)$, $B_{2}\left(f\right)=D_{1}\left(f\right)-D_{3}\left(f\right)$, $B_{3}\left(f\right)=-\left(D_{2}\left(f\right)+D_{3}\left(f\right)\right)$, and $B_{4}\left(f\right)=D_{2}\left(f\right)-D_{3}\left(f\right)$.

Next, we analyse the performance of the response of the two ears. In [28], the distance *d* between two ears is approximately equal to 1.2 mm. The incident plane wave impinges on the two ears from ϑ, where ϑ is the angle between the propagation direction and the normal of bilateral tympanal membranes. It is assumed that the parameters of each ear are the same, viz., $\beta_{1}=\beta_{2}$ and $\alpha_{1}=\alpha_{2}$. The values of parameters $\beta_{i}$, $\alpha_{i}$, and $m_{0}$ (i=1,⋯,3) can be found in [28]. Taking the centre of two ears as the reference point, the response of the two ears can be obtained:(8)$$\begin{array}{c} H_{1}\left(f\right)=\frac{j\sin(2\pi f\tau /2)}{B_{2}\left(f\right)}+\frac{\cos(2\pi f\tau /2)}{B_{1}\left(f\right)},\\ H_{2}\left(f\right)=-\frac{j\sin(2\pi f\tau /2)}{B_{2}\left(f\right)}+\frac{\cos(2\pi f\tau /2)}{B_{1}\left(f\right)}, \end{array}$$
where $\tau =d\cos\left(\vartheta \right)/c_{c}$ with $c_{c}\approx 344\;m/s$. To demonstrate the effect of the coupling system of the *Ormia*’s ears, Figure 3 shows the biauricular amplitude difference of the coupling and uncoupling response ($\beta_{3}=0$ and $\alpha_{3}=0$); meanwhile, Figure 4 shows the biauricular phase difference of the coupling and uncoupling response ($\beta_{3}=0$ and $\alpha_{3}=0$). It can be seen from Figure 3 and Figure 4 that the mechanical coupling system amplifies the amplitude and phase differences between the responses of the two ears. It is demonstrated that the mechanical coupling system can effectively create a larger distance between the two ears.

## 3. Proposed Method

In this section, we first introduce the mechanical coupling system of the *Ormia*’s ears into the small-sized BCAVSA to obtain a virtual BCAVSA with a larger aperture. Then, based on the principle of the harmonic nesting, we propose a frequency invariant beamforming for the virtual BCAVSA to form the beampattern with the constant beamwidth on the azimuth and elevation planes.

### 3.1. Coupling Magnified BCAVSA

According to Equation (Equation 3), the steering vectors of the two-element pressure sensor array and the uniform circular pressure sensor array are two parts of constituting the steering vector of the *q*th cylindrical AVSA. Consequently, we first consider extending the magnification effect of the coupling system of the *Ormia*’s ears to a two-element pressure sensor array. The two pressure sensors are identical, and they have the same spring $\alpha_{1}=\alpha_{2}=\alpha_{0}$ and the same dash-pot $\beta_{1}=\beta_{2}=\beta_{0}$. According to Equation (Equation 7), we can obtain the coupling magnified matrix for the two-element pressure sensor array, and it can be expressed as:(9)$$T_{l}\left(f\right)=H_{0}^{-1}\left(f\right)\Lambda_{0},$$
where $\Lambda_{0}=\left[\begin{array}{cc} 1 & -1\\ 1 & 1 \end{array}\right]$, $H_{0}\left(f\right)=\left[\begin{array}{cc} \Gamma (f) & -\Gamma (f)\\ \Upsilon (f) & \Upsilon (f) \end{array}\right]$ with $\Gamma \left(f\right)=\alpha_{0}+2\alpha_{3}+j2\pi f\left(\beta_{0}+2\beta_{3}\right)-m_{0}{\left(2\pi f\right)}^{2}$ and $\Upsilon \left(f\right)=\beta_{0}+j2\pi f\alpha_{0}-m_{0}{\left(2\pi f\right)}^{2}$. ${\left(\;\cdot \;\right)}^{-1}$ denotes the inverse operation.

We next extend the magnification effect of the coupling system of the *Ormia ochracea*’s ears to the uniform circular pressure sensor array composed of *N* identical pressure sensors which own the same spring $\alpha_{0}$ and the same dash-pot $\beta_{0}$. Since a coupling magnification can be constructed among the two successive sensors, the *N* circular pressure sensor array can form *N* coupling magnifications. According to the analysis, the coupling magnified matrix for the circular array can be expressed as follows:(10)$$T_{c}\left(f\right)=H_{c}^{-1}\left(f\right)\Lambda_{c},$$
where
(11)$$\begin{array}{c} H_{c}^{-1}\left(f\right)={\left[\begin{array}{cc} H_{0}\left(f\right) & \;\\ \ddots \\ \; & H_{0}\left(f\right) \end{array}\right]}^{-1},\\ \Lambda_{c}=I_{N}⊗\left[\begin{array}{c} 1\\ 1 \end{array}\right]+{\bar{I}}_{N}⊗\left[\begin{array}{c} -1\\ 1 \end{array}\right], \end{array}$$
where $H_{c}^{-1}\left(f\right)$ and $\Lambda_{c}$ are the 2N×2N and 2N×N matrices, respectively. $I_{N}$ is the N×N identity matrix. ${\bar{I}}_{N}=\left[\begin{array}{cc} 0_{(N-1)\times 1} & I_{\left(N-1\right)}\\ 1 & 0_{(N-1)\times 1}^{T} \end{array}\right]$, where $0_{(N-1)\times 1}$ is the (N−1)×1 zero vector.

According to Equations (Equation 9) and (Equation 10), the coupling magnified matrix for the *q*th cylindrical AVSA composed of the −qth and *q*th UCAVSAs can be deduced as (q=1,t⋯,Q):(12)$$T_{\mp q}\left(f\right)=T_{l,q}\left(f\right)⊗T_{c,q}\left(f\right)⊗I_{4},$$
where $T_{l,q}\left(f\right)$ is the coupling magnified matrix corresponding to the *q*th two-element pressure sensor array, and $T_{c,q}\left(f\right)$ is the coupling magnified matrix for the *q*th the uniform circular pressure sensor array. $I_{4}$ is the 4×4 identity matrix.

According to Equation (Equation 12), the coupling magnified matrix for the BCAVSA can be obtained, and it can be expressed as:(13)$$T\left(f\right)={\left[T_{\mp 1}\left(f\right),\cdots ,T_{\mp Q}\left(f\right)\right]}^{T}.$$

Consequently, we can apply the coupling magnified matrix T(f) to the BCAVSA to obtain a virtual BCAVSA with the larger interelement spacing. The steering vector of the virtual BCAVSA can be expressed as:(14)$${\tilde{a}}_{v}\left(f,\theta_{s},\phi_{s}\right)=T\left(f\right)a_{v}\left(f,\theta_{s},\phi_{s}\right)={\left[{\tilde{a}}_{v,\mp 1}\left(f,\theta_{s},\phi_{s}\right),\cdots ,{\tilde{a}}_{v,\mp Q}\left(f,\theta_{s},\phi_{s}\right)\right]}^{T},$$
where ${\tilde{a}}_{v,\mp q}\left(f,\theta_{s},\phi_{s}\right)=T_{\mp q}\left(f\right)a_{v,\mp q}\left(f,\theta_{s},\phi_{s}\right)={\tilde{a}}_{z,q}\left(f,\phi_{s}\right)⊗{\tilde{a}}_{pc,q}\left(f,\theta_{s},\phi_{s}\right)⊗u\left(\theta_{s},\phi_{s}\right)$ with ${\tilde{a}}_{z,q}\left(f,\phi_{s}\right)=T_{l,q}\left(f\right)a_{z,q}\left(f,\phi_{s}\right)$ and ${\tilde{a}}_{pc,q}\left(f,\theta_{s},\phi_{s}\right)=T_{c,q}\left(f\right)a_{pc,q}\left(f,\theta_{s},\phi_{s}\right)$.

It can be seen from Equation (Equation 14) that the performance of the coupling system relies on the coupling parameters. In order to obtain the good effect of the phase difference enhancement for the passive sonar system, it is necessary to set the parameters of the coupling system reasonably. According to [28], it can be known that the *Ormia ochracea* are sensitive to the sound signal with $f_{0}=5\;\mathrm{kHz}$ from the cricket host and the distance between the *Ormia ochracea*’s two ears is approximately equal to $d_{0}=1.2\;\mathrm{mm}$. When the coupling processing is performed on the signal with frequency *f*, the parameter $m_{0}$ remains unchanged and the parameters $\beta_{i}\left(i=0,3\right)$ and $\alpha_{i}\left(i=0,3\right)$ in the matrix $T_{c,q}\left(f\right)$ can be set to be $\left[f\sqrt[{4}]{d_{c,q}}/\left(f_{0}\sqrt{d_{0}}\right)\right]$ and ${\left[f\sqrt[{4}]{d_{c,q}}/\left(f_{0}\sqrt{d_{0}}\right)\right]}^{2}$ times of the original parameters to obtain the phase difference enhancement of $a_{pc,q}\left(f,\theta_{s},\phi_{s}\right)$, where $d_{c,q}=2r_{q}\sin\left(\pi /N\right)$ is the distance of the two adjacent AVSs of the *q*th UCAVSA, q=1,⋯,Q. Similarly, in order to obtain the phase difference enhancement for $a_{z,q}\left(f,\phi_{s}\right)$ at the signal frequency *f*, the parameters $\beta_{i}\;\left(i=0,3\right)$ and $\alpha_{i}\;\left(i=0,3\right)$ in the matrix $T_{l,q}\left(f\right)$ can be set to be $\left(f\sqrt[{4}]{h_{q}}/\left(f_{0}\sqrt{d_{0}}\right)\right)$ and ${\left(f\sqrt[{4}]{h_{q}}/\left(f_{0}\sqrt{d_{0}}\right)\right)}^{2}$ times of the original parameters, where $f_{0}=5\;\mathrm{kHz}$ and $d_{0}=1.2\;\mathrm{mm}$ are the sound signal from the cricket host and the distance between the *Ormia ochracea*’s two ears, respectively.

### 3.2. Frequency Invariant Beamforming Method

We develop the harmonic nesting method to the virtual BCAVSA to achieve a desired frequency invariant beampattern in the azimuth and elevation planes. First, we take an easy example to understand the harmonic nesting method. We take two uniform linear arrays as the example. It is assumed that the high-frequency and the low-frequency uniform linear arrays have the same number of elements, and the interelement spacing of the high-frequency array is half of the low-frequency array. It is assumed that the desired broadband is equal to $\left[f_{l1},f_{l2}\right]$, where $f_{l2}=2f_{l1}$. At any frequency $f\in [f_{l1},f_{l2}]$, the directivity functions $P_{L}\left(f\right)$ and $P_{H}\left(f\right)$ for the low-frequency and high-frequency linear arrays can be calculated. The weighted coefficients $R_{L}\left(f\right)$ and $R_{H}\left(f\right)$ can be calculated by using the directivity functions. Then, the $R_{L}\left(f\right)$ and $R_{H}\left(f\right)$ are used to multiply the directivity functions $P_{L}\left(f\right)$ and $P_{H}\left(f\right)$, respectively. Finally, summing $R_{L}\left(f\right)P_{L}\left(f\right)$ and $R_{H}\left(f\right)P_{H}\left(f\right)$, the frequency invariant beamforming V(f) can be obtained. The specific implementation process is shown in Figure 5.

Then, we introduce the above-mentioned harmonic nesting method into the virtual BCAVSA to form the constant beamwidth beampattern. The directivity pattern of the *q*th cylindrical AVSA of the virtual BCAVSA under the signal frequency *f* can be obtained by setting the weighted vector for the *q*th cylindrical AVSA equal to the normalized steering vector, and it can be written as:(15)$$\begin{array}{c} w_{v,\mp q}\left(f,\theta_{s},\phi_{s}\right)=\frac{{\tilde{a}}_{v,\mp q}\left(f,\theta_{s},\phi_{s}\right)}{{\tilde{a}}_{v,\mp q}^{H}\left(f,\theta_{s},\phi_{s}\right){\tilde{a}}_{v,\mp q}\left(f,\theta_{s},\phi_{s}\right)}\\ ={\bar{a}}_{z,q}\left(f,\phi_{s}\right)⊗{\bar{a}}_{pc,q}\left(f,\theta_{s},\phi_{s}\right)⊗\bar{u}\left(\theta_{s},\phi_{s}\right), \end{array}$$
where ${\bar{a}}_{z,q}\left(f,\phi_{s}\right)={\tilde{a}}_{z,q}\left(f,\phi_{s}\right)/\left[{{\tilde{a}}_{z,q}}^{H}\left(f,\phi_{s}\right){\tilde{a}}_{z,q}\left(f,\phi_{s}\right)\right]$, ${\bar{a}}_{pc,q}\left(f,\theta_{s},\phi_{s}\right)={\tilde{a}}_{pc,q}\left(f,\theta_{s},\phi_{s}\right)/\left[{{\tilde{a}}_{pc,q}}^{H}\left(f,\theta_{s},\phi_{s}\right){\tilde{a}}_{pc,q}\left(f,\theta_{s},\phi_{s}\right)\right]$, and $\bar{u}\left(\theta_{s},\phi_{s}\right)=u(\theta_{s},\phi_{s})/\left[u^{H}\left(\theta_{s},\phi_{s}\right)u\left(\theta_{s},\phi_{s}\right)\right]$, and $u^{H}\left(\theta_{s},\phi_{s}\right)u\left(\theta_{s},\phi_{s}\right)=2$.

For the *q*th cylindrical AVSA of the virtual BCAVSA, the directivity function associated with the signal frequency *f* can be written as:(16)$$\begin{array}{c} {\overset{˘}{P}}_{q}\left(f,\theta ,\phi \right)=w_{v,\mp q}^{H}\left(f,\theta_{s},\phi_{s}\right){\tilde{a}}_{v,\mp q}\left(f,\theta ,\phi \right)\\ ={\overset{˘}{P}}_{z,q}\left(f,\phi \right){\overset{˘}{P}}_{c,q}\left(f,\theta ,\phi \right){\overset{˘}{P}}_{u}\left(\theta ,\phi \right), \end{array}$$
where ${\overset{˘}{P}}_{z,q}\left(f,\phi \right)={\bar{a}}_{z,q}^{H}\left(f,\phi_{s}\right){\tilde{a}}_{z,q}\left(f,\phi \right)=\cos\left[kz_{q}\left(\cos\phi -\cos\phi_{s}\right)\right]$, ${\overset{˘}{P}}_{c,q}\left(f,\theta ,\phi \right)={\bar{a}}_{pc,q}^{H}\left(f,\theta_{s},\phi_{s}\right){\tilde{a}}_{pc,q}\left(f,\theta ,\phi \right)$, and ${\overset{˘}{P}}_{u}\left(\theta ,\phi \right)={\bar{u}}^{H}\left(\theta_{s},\phi_{s}\right)u\left(\theta ,\phi \right)$, which is independent from the signal frequency. From Equation (Equation 16), it can be seen that the directivity function of the *q*th cylindrical AVSA can be treated as the product of three directivity functions which are respectively corresponding to the two-element pressure sensor array along the *z*-axis, the uniform circular sensor arrays, and a single acoustic vector sensor. The normalized directivity function of the *q*th cylindrical AVSA can be written as:(17)$$\begin{array}{c} P_{q}\left(f,\theta ,\phi \right)=\frac{|{\overset{˘}{P}}_{q}\left(f,\theta ,\phi \right)|}{\max\{|{\overset{˘}{P}}_{q}\left(f,\theta ,\phi \right)|\}}\\ =\frac{|{\overset{˘}{P}}_{z,q}\left(f,\phi \right){\overset{˘}{P}}_{c,q}\left(f,\theta ,\phi \right){\overset{˘}{P}}_{u}\left(\theta ,\phi \right)|}{\left|\max\{{\overset{˘}{P}}_{z,q}\left(f,\phi \right){\overset{˘}{P}}_{c,q}\left(f,\theta ,\phi \right){\overset{˘}{P}}_{u}\left(\theta ,\phi \right)\}\right|}\\ =\frac{|{\overset{˘}{P}}_{z,q}\left(f,\phi \right)|}{\left|\max\{{\overset{˘}{P}}_{z,q}\left(f,\phi \right)\}\right|}\frac{|{\overset{˘}{P}}_{c,q}\left(f,\theta ,\phi \right)|}{\left|\max\{{\overset{˘}{P}}_{c,q}\left(f,\theta ,\phi \right)\}\right|}\frac{|{\overset{˘}{P}}_{u}\left(\theta ,\phi \right)|}{\left|\max\{{\overset{˘}{P}}_{c,q}\left(f,\theta ,\phi \right)\}\right|}\\ =P_{z,q}\left(f,\phi \right)P_{c,q}\left(f,\theta ,\phi \right)P_{u}\left(\theta ,\phi \right), \end{array}$$
where max{(·)} denotes the maximum value of the function (·). $P_{z,q}\left(f,\phi \right)=|{\overset{˘}{P}}_{z,q}\left(f,\phi \right)|/\max\{|{\overset{˘}{P}}_{z,q}\left(f,\phi \right)\left|\right\}$, $P_{c,q}\left(f,\theta ,\phi \right)=|{\overset{˘}{P}}_{c,q}\left(f,\theta ,\phi \right)|/\max\{|{\overset{˘}{P}}_{c,q}\left(f,\theta ,\phi \right)\left|\right\}$, and $P_{u}\left(\theta ,\phi \right)=|{\overset{˘}{P}}_{u}\left(\theta ,\phi \right)|/\max\{|{\overset{˘}{P}}_{u}\left(\theta ,\phi \right)\left|\right\}$.

The directivity function of the *q*th cylindrical AVSA is the function of the signal frequency *f*. As the signal frequency increases, the beamwidth of the directivity function of the *q*th cylindrical AVSA becomes narrower. When the AVSA receives the broadband signal from the volumetric element, the constant beamwidth pattern is quite important to obtain the signal energy of the radiated volumetric element exactly. In order to obtain the constant beamwidth beampattern in the azimuth and elevation planes, the harmonic nesting method is extended into this paper to form the constant beamwidth pattern.

We take an octave $\left[f_{L},\;f_{H}\right]$ as an example, where $f_{H}=2f_{L}$. As shown in Figure 1, the *q*th cylindrical AVSA is composed of the −qth and *q*th UCAVSAs (q=1,2). According to Equation (Equation 17), the expressions of the directivity function of the high-frequency and low-frequency cylindrical AVSA can be written as:(18)$$\begin{array}{c} P_{H}\left(f,\theta ,\phi \right)=P_{z,1}\left(f,\phi \right)P_{c,1}\left(f,\theta ,\phi \right)P_{u}\left(\theta ,\phi \right),\\ P_{L}\left(f,\theta ,\phi \right)=P_{z,2}\left(f,\phi \right)P_{c,2}\left(f,\theta ,\phi \right)P_{u}\left(\theta ,\phi \right). \end{array}$$

We set the radius of the high-frequency cylindrical AVSA is equal to half that of the low-frequency cylindrical AVSA, viz., $2r_{1}=r_{2}$. Meanwhile, the height of the high-frequency cylindrical AVSA is equal to half that of the low-frequency cylindrical AVSA, viz., $2z_{1}=z_{2}$. In addition, the norm of the difference between the coupling magnified matrix $T_{\mp 1}\left(f_{H}\right)$ associated with the high-frequency cylindrical AVSA and the coupling magnified matrix $T_{\mp 2}\left(f_{L}\right)$ associated with the low-frequency cylindrical AVSA is equal to $∥T_{\mp 1}\left(f_{H}\right)-T_{\mp 2}\left(f_{L}\right){∥}_{2}$, and it is close to zero. Consequently, the directivity function of the high-frequency cylindrical AVSA associated with the $f_{H}$ is approximately equal to that of the low-frequency cylindrical AVSA corresponding to the $f_{L}$:(19)$$P_{H}\left(f_{H},\theta ,\phi \right)=P_{L}\left(f_{L},\theta ,\phi \right).$$

Since $P_{u}\left(\theta ,\phi \right)$ is independent from the signal frequency, it can be deduced from Equation (Equation 19):(20)$$\begin{array}{cc} P_{z,1}\left(f_{H},\phi \right) & =P_{z,2}\left(f_{L},\phi \right),\\ P_{c,2}\left(f_{H},\theta ,\phi \right) & =P_{c,2}\left(f_{L},\theta ,\phi \right). \end{array}$$

To maintain a constant beamwidth of the high- and low-frequency cylindrical AVSAs under the bandwidth $\left[f_{L},f_{H}\right]$, the frequency weighted coefficients are necessary to introduce into the two cylindrical AVSAs. Linear combination of the directivity functions of the two cylindrical AVSAs using the frequency weighted coefficients can obtain the directivity function with a constant beamwidth under the bandwidth $\left[f_{L},f_{H}\right]$. In order to obtain the frequency weighted coefficients for the two cylindrical AVSAs with different interelement spacing, we will consider the azimuth and elevation planes separately.

Firstly, to form the constant beampattern in the azimuth plane, the directivity function can be obtained by doing the following transformation, and it can be expressed as:(21)$$\begin{array}{cc} V_{1}\left(f,\theta ,\phi \right) & =R_{c,1}\left(f\right)P_{H}\left(f,\theta ,\phi \right)\\ & +R_{c,2}\left(f\right)\Gamma_{z}\left(f,\phi \right)P_{L}\left(f,\theta ,\phi \right)\\ & =V_{c}\left(f,\theta ,\phi \right)P_{z,1}\left(f,\phi \right)P_{u}\left(\theta ,\phi \right), \end{array}$$
where $\Gamma_{z}\left(f,\phi \right)=P_{z,1}\left(f,\phi \right)/P_{z,2}\left(f,\phi \right)$. $R_{c,1}\left(f\right)$ and $R_{c,2}\left(f\right)$ are the frequency weighted coefficients associated with the high- and low-frequency cylindrical AVSAs. $V_{c}\left(f,\theta ,\phi \right)=R_{c,1}\left(f\right)P_{c,1}\left(f,\theta ,\phi \right)+R_{c,2}\left(f\right)P_{c,2}\left(f,\theta ,\phi \right)$. For the azimuth plane, we set $\phi =\phi_{s}$. Due to $P_{z,1}\left(f,\phi_{s}\right)=1$. Equation (Equation 21) can be simplified as:(22)$$\begin{array}{c} V_{1}\left(f,\theta ,\phi_{s}\right)=V_{c}\left(f,\theta ,\phi_{s}\right)P_{u}\left(\theta ,\phi_{s}\right). \end{array}$$

We need two functions to solve the frequency weighted coefficients $R_{c,1}\left(f\right)$ and $R_{c,2}\left(f\right)$. Since $P_{u}\left(\theta ,\phi_{s}\right)$ is independent from the signal frequency, we only calculate the frequency weighted coefficients using the directivity function $V_{c}\left(f,\theta ,\phi_{s}\right)$. According to [25], we can select the main axis and the half-power point of the normalized directivity function $V_{c}\left(f,\theta ,\phi_{s}\right)$ to calculate the $R_{c,1}\left(f\right)$ and $R_{c,2}\left(f\right)$:

(1) when we set $\theta =\theta_{s}$, the normalized value of $V_{c}\left(f,\theta ,\phi \right)$ is equal to 1, and it can be written as:(23)$$|R_{c,1}\left(f\right)+R_{c,2}\left(f\right)|=1$$
viz., it means that the frequency response in the octave $\left[f_{L},f_{H}\right]$ is flat and the amplitude is equal to 1.

(2) when $\theta =\theta_{s}+\delta_{\theta }$ ($\delta_{\theta }={BW}_{c,0.5}/2$, ${BW}_{c,0.5}$ is of the beamwidth of the half-power point), the following can be obtained:(24)$$\left|\frac{V_{c}\left(f,\theta_{s}+\delta_{\theta },\phi_{s}\right)}{V_{c}\left(f,\theta_{s},\phi_{s}\right)}\right|=\frac{1}{\sqrt{2}}.$$

Due to $P_{c,1}\left(f_{H},\theta_{s},\phi_{s}\right)=P_{c,2}\left(f_{L},\theta_{s},\phi_{s}\right)=1$, by solving Equations (Equation 23) and (Equation 24), the frequency weighted coefficients can be obtained, and they can written as:(25)$$\begin{array}{c} R_{c,1}\left(f\right)=\frac{1-\sqrt{2}P_{c,2}\left(f,\theta_{s}+\delta_{\theta },\phi_{s}\right)}{\sqrt{2}\left[P_{c,1}\left(f,\theta_{s}+\delta_{\theta },\phi_{s}\right)-P_{c,2}\left(f,\theta_{s}+\delta_{\theta },\phi_{s}\right)\right]},\\ R_{c,2}\left(f\right)=\frac{\sqrt{2}P_{c,1}\left(f,\theta_{s}+\delta_{\theta },\phi_{s}\right)-1}{\sqrt{2}\left[P_{c,1}\left(f,\theta_{s}+\delta_{\theta },\phi_{s}\right)-P_{c,2}\left(f,\theta_{s}+\delta_{\theta },\phi_{s}\right)\right]}. \end{array}$$

Next, we consider the frequency invariant directivity function in the elevation plane. To obtain $P_{z,2}\left(f,\phi \right)$, $V_{1}\left(f,\theta ,\phi \right)$ in Equation (Equation 21) can be converted into:(26)$$\begin{array}{cc} V_{2}\left(f,\theta ,\phi \right) & =\frac{V_{1}\left(f,\theta ,\phi \right)}{\Gamma_{z}\left(f,\phi \right)}\\ & =V_{c}\left(f,\theta ,\phi \right)P_{z,2}\left(f,\phi \right)P_{u}\left(\theta ,\phi \right). \end{array}$$

In $V_{1}\left(f,\theta ,\phi \right)$ and $V_{2}\left(f,\theta ,\phi \right)$, $V_{c}\left(f,\theta ,\phi \right)$ and $P_{u}\left(\theta ,\phi \right)$ are equivalent; meanwhile, $P_{z,1}\left(f,\phi \right)$ and $P_{z,2}\left(f,\phi \right)$ are the directivity functions associated with the high- and low-frequency scalar two-element arrays, respectively. Consequently, there are a couple of the frequency weighted coefficients to make the directivity function maintain a constant beamwidth in the elevation plane, and the directivity function can be expressed as:(27)$$\begin{array}{cc} V_{b}\left(f,\theta ,\phi \right) & =R_{l,1}\left(f\right)V_{1}\left(f,\theta ,\phi \right)+R_{l,2}\left(f\right)V_{2}\left(f,\theta ,\phi \right)\\ & =V_{c}\left(f,\theta ,\phi \right)V_{l}\left(f,\phi \right)P_{u}\left(\theta ,\phi \right), \end{array}$$
where $R_{l,1}\left(f\right)$ and $R_{l,2}\left(f\right)$ are the frequency weighted coefficients for the the high- and low-frequency scalar two-element arrays, respectively. $V_{l}\left(f,\phi \right)=R_{l,1}\left(f\right)P_{z,1}\left(f,\phi \right)+R_{l,2}\left(f\right)P_{z,2}\left(f,\phi \right)$.

Similar to Equations (Equation 23) and (Equation 24), the frequency weighted coefficients $R_{l,1}\left(f\right)$ and $R_{l,2}\left(f\right)$ among the matrix $V_{l}\left(f,\phi \right)$ can be solved by:(28)$$\begin{array}{cc} |R_{l,1}\left(f\right)+R_{l,2}\left(f\right)| & =1,\\ \left|\frac{V_{l}\left(f,\phi_{s}+\delta_{\phi }\right)}{V_{l}\left(f,\phi_{s}\right)}\right| & =\frac{1}{\sqrt{2}}, \end{array}$$
where $\delta_{\phi }=BW_{l,0.5}/2$, $BW_{l,0.5}$ is the beamwidth at the half power point. By solving Equation (Equation 28) $R_{l,1}\left(f\right)$ and $R_{l,2}\left(f\right)$ are equal to:(29)$$\begin{array}{c} R_{l,1}\left(f\right)=\frac{1-\sqrt{2}P_{z,2}\left(f,\phi_{s}+\delta_{\phi }\right)}{\sqrt{2}\left[P_{z,1}\left(f,\phi_{s}+\delta_{\phi }\right)-P_{z,2}\left(f,\phi_{s}+\delta_{\phi }\right)\right]},\\ R_{l,2}\left(f\right)=\frac{\sqrt{2}P_{z,1}\left(f,\phi_{s}+\delta_{\phi }\right)-1}{\sqrt{2}\left[P_{z,1}\left(f,\phi_{s}+\delta_{\phi }\right)-P_{z,2}\left(f,\phi_{s}+\delta_{\phi }\right)\right]}. \end{array}$$

According to Equations (Equation 21), (Equation 25), (Equation 26), (Equation 27), and (Equation 29), the frequency weighted coefficients for the high- and low-frequency cylindrical AVSAs $P_{H}\left(f,\theta ,\phi \right)$ and $P_{L}\left(f,\theta ,\phi \right)$ can be obtained:(30)$$\begin{array}{c} V_{b}\left(f,\theta ,\phi \right)=R_{l,1}\left(f\right)V_{1}\left(f,\theta ,\phi \right)+R\left(l,2\right)V_{2}\left(f,\theta ,\phi \right)\\ =R_{l,1}\left(f\right)\left\{R_{c,1}\left(f\right)P_{H}\left(f,\theta ,\phi \right)+R_{c,2}\left(f\right)\Gamma \left(f,\phi \right)P_{L}\left(f,\theta ,\phi \right)\right\}\\ +R_{l,2}\left(f\right)\left\{R_{c,1}\left(f\right)P_{H}\left(f,\theta ,\phi \right)/\Gamma \left(f,\phi \right)+R_{c,2}\left(f\right)P_{L}\left(f,\theta ,\phi \right)\right\}\\ =\left[R_{l,1}\left(f\right)R_{c,1}\left(f\right)+\frac{R_{l,2}\left(f\right)R_{c,1}\left(f\right)}{\Gamma (f,\phi )}\right]P_{H}\left(f,\theta ,\phi \right)\\ +\left[R_{l,1}\left(f\right)R_{c,2}\left(f\right)\Gamma \left(f,\phi \right)+R_{l,2}\left(f\right)R_{c,2}\left(f\right)\right]P_{L}\left(f,\theta ,\phi \right)\\ =R_{H}\left(f,\phi \right)P_{H}\left(f,\theta ,\phi \right)+R_{L}\left(f,\phi \right)P_{L}\left(f,\theta ,\phi \right), \end{array}$$
where $R_{H}\left(f,\phi \right)=R_{l,1}\left(f\right)R_{c,1}\left(f\right)+R_{l,2}\left(f\right)R_{c,1}\left(f\right)/\Gamma (f,\phi )$ and $R_{L}\left(f,\phi \right)=R_{l,1}\left(f\right)R_{c,2}\left(f\right)\Gamma \left(f,\phi \right)+R_{l,2}\left(f\right)R_{c,2}\left(f\right)$. It can be seen from Equation (Equation 30) that, by using the frequency weighted coefficients $R_{H}\left(f,\phi \right)$ and $R_{L}\left(f,\phi \right)$ to combine $P_{H}\left(f,\theta ,\phi \right)$ and $P_{L}\left(f,\theta ,\phi \right)$ linearly, the directivity function $V_{b}\left(f,\theta ,\phi \right)$ is obtained, which has constant beamwidth pattern in azimuth and elevation planes.

Although the above-described theory and method are designed for the two cylindrical AVSAs to form the constant beamwidth over the octave, this constant beamwidth beamforming method can be extended to the multiple cylindrical AVSAs to construct the frequency invariant beampattern over multiple octaves. Consider *Q* cylindrical AVSAs, where the radii and heights of the 1st to the *Q*th cylindrical AVSAs are equal to $r_{1},2r_{2},\cdots ,2^{Q-1}r_{1}$ and $h_{1},2h_{1},\cdots ,2^{Q-1}h_{1}$ ($h_{1}=2z_{1}$), respectively. It is assumed that the broadband is equal to $\left[f_{L},f_{H}\right]$, where $f_{L}=f_{H}/2^{Q-1}$, which can be divided into Q−1 segments, and there are $\left[f_{H}/2^{Q-1},f_{H}/2^{Q-2}\right],\cdots ,\left[f_{H}/4,f_{H}/2\right],\left[f_{H}/2,f_{H}\right]$. For the $f\in [f_{H}/2^{Q-1},f_{H}/2^{Q-2}]$, the directivity functions $P_{H,1}\left(f,\theta ,\phi \right)$ and $P_{L,1}\left(f,\theta ,\phi \right)$ for the 1st and 2nd cylindrical AVSAs can be calculated by Equation (Equation 18); similarly, the frequency weighted coefficients $R_{H,1}\left(f,\phi \right)$ and $R_{L,1}\left(f,\phi \right)$ can be also calculated by Equation (Equation 30). Then, by summing $R_{H,1}\left(f,\phi \right)P_{H,1}\left(f,\theta ,\phi \right)$ and $R_{L,1}\left(f,\phi \right)P_{L,1}\left(f,\theta ,\phi \right)$, the constant beamwidth pattern over the $\left[f_{H}/2^{Q-1},f_{H}/2^{Q-2}\right]$ can be obtained. Similarly, using the *q*th and (q+1)th cylindrical AVSAs, the directivity functions $P_{H,q}\left(f,\theta ,\phi \right)$ and $P_{L,q}\left(f,\theta ,\phi \right)$ and weighted frequency coefficients $R_{H,q}\left(f,\phi \right)$ and $R_{L,q}\left(f,\phi \right)$ can be calculated based on Equations (Equation 18) and (Equation 30). They can form the frequency invariant beampattern over the octave $\left[f_{H}/2^{q},f_{H}/2^{q-1}\right]$ by using $R_{H,q}\left(f,\phi \right)P_{H,q}\left(f,\theta ,\phi \right)+R_{L,q}\left(f,\phi \right)P_{L,q}\left(f,\theta ,\phi \right)$, which has the same beamwidth to that of using the 1st and 2nd cylindrical AVSAs. Finally, the frequency invariant beampattern over $\left[f_{L},f_{H}\right]$ can be obtained by combining Q−1 constant beamwidth pattern. The schematic diagram of forming a constant beamwidth over Q−1 octave using a BCAVSA composed of *Q* cylindrical AVSAs is shown in Figure 6.

Finally, the array gain is discussed in this section. It is assumed that the signal is a unidirectional plane wave and the noise is isotropic. That is, the signal is perfectly coherent and the noise power in per unit solid angle is the same in all directions. At this time, the array gain reduces to the quantity called directivity index. In the case of a plane-wave signal in the isotropic noise impinging on the array and the BCAVSA steered in the direction of the signal, the array gain expression using the directivity function of the signal and noise can be written as [40]:(31)$$\begin{array}{c} AG=DI=10\mathrm{lg}\frac{\int_{4\pi }d\Omega }{\int_{4\pi }V_{b}\left(f,\theta ,\phi \right)d\Omega }\\ =10\mathrm{lg}\frac{4\pi }{\int_{0}^{2\pi }\int_{0}^{\pi }V_{b}\left(f,\theta ,\phi \right)\;\sin\phi \;d\theta d\phi }, \end{array}$$
where AG and DI denote the array gain and the directivity index, respectively. Ω denotes the unit solid angle.

## 4. Simulation Results

In this section, we provide some computational simulations to illustrate the frequency invariant property of the proposed method and the advantages of the proposed method compared to the original BCAVSA and the bi-cone pressure sensor array with the same configuration parameters. The BCAVSA is composed of several groups of cylindrical AVSAs. Each cylindrical AVSA consists of two UCAVSAs, and each UCAVSA is composed of eight AVSs. The directions of the acoustic particle velocity components of Each AVS is oriented along the *x*-, *y*-, and *z*-axes.

### 4.1. Original BCAVSA and Coupling Magnified BCAVSA

We consider a BCAVSA composed of two cylindrical AVSAs. The desired bandwidth is equal to f∈[500 Hz,1000 Hz]. The radii of the first and second cylindrical AVSA are equal to $r_{1}=0.225\;m=0.15\lambda_{H}$ and $r_{2}=0.45\;m=0.3\lambda_{H}$, respectively, $\lambda_{H}=c/f_{H}$ is the wavelength associated with $f_{H}$. The two UCAVSAs among the first cylindrical AVSA respectively locates at $z_{1}=0.075\;m=0.05\lambda_{H}$ and $z_{-1}=-0.075\;m=-0.05\lambda_{H}$. The two UCAVSAs among the second cylindrical AVSA respectively locates at $z_{2}=0.15\;m=0.1\lambda_{H}$ and $z_{-2}=-0.15\;m=-0.1\lambda_{H}$. The sound speed in the underwater is equal to 1500 m/s. The main beam is targeted at $\theta_{s}=180^{\circ }$ and $\phi_{s}=90^{\circ }$. As the beampattern is a function of three variables *f*, θ, and ϕ, we only provide the beampatterns at some frequency bins. Figure 7 shows the normalized beampatterns of the coupling magnified and the original BCAVSAs at f=500 Hz and 700 Hz. Figure 8 shows the beampatterns in the azimuth plane, which correspond to the coupling magnified and the original BCAVSAs, respectively, where the elevation angle is equal to $\phi =90{}^{\circ }$. Figure 9 shows the beampatterns in the elevation plane, which correspond to the coupling magnified and the original BCAVSAs, where the azimuth angle is $\theta =180{}^{\circ }$.

It can be seen from Figure 7 that the original and coupling magnified BCAVSAs can both form the constant beamwidth patterns at 500 Hz and 700 Hz. In addition, the harmonic nesting method based on the coupling magnified BCAVSA has a much narrower mainlobe and lower sidelobes than that using the original BCAVSA. The frequency invariant property of the proposed constant beamwidth beamforming method can be again clearly discerned from Figure 8 and Figure 9. In Figure 8 and Figure 9, the frequency invariant beampattern of the coupling magnified BCAVSA has a 66° beamwidth in the azimuth plane and a 20° beamwidth in the elevation plane, while the frequency invariant beampattern of the original BCAVSA has a 98° beamwidth in the azimuth plane and a 94° beamwidth in the elevation plane. From Figure 7, Figure 8 and Figure 9, we know that the coupling magnification processing inspired by the *Ormia ochracea*’s ears can be applied in the BCAVSA to increase the interelement distance. As a result, compared to the original BCAVSA, the coupling magnified BCAVSA has lower sidelobes and a narrower mainlobe. According to Equation (Equation 31), if the mainlobe is narrower and sidelobes is lower, the denominator in Equation (Equation 31) is smaller, and thus the AG is higher. Consequently, the coupling magnified BCAVSA has a higher AG than the original BCAVSA.

### 4.2. Coupling Magnified BCAVSA and a Bi-Cone Pressure Sensor Array

Except for the main beam targeted at $\theta_{s}=130^{\circ }$ and $\phi_{s}=90^{\circ }$, the other simulation conditions are the same as those in Section 4.1. The bi-cone pressure sensor array has the same configuration parameters as the BCAVSA. Figure 10 shows the beampatterns of the coupling magnified BCAVSA and bi-cone pressure sensor array in the azimuth plane, where the elevation angle is fixed to 90°. Figure 11 shows the beampatterns of the coupling magnified BCAVSA and bi-cone pressure sensor array in the elevation plane, where the azimuth angle is fixed to 130°.

It can be seen from Figure 10 that the beampattern of the coupling magnified BCAVSA in the azimuth plane has the lower sidelobes and a narrower mainlobe compared to those of the coupling magnified bi-cone pressure sensor array. This phenomenon is caused by the inherent characteristic of the AVSA, viz., the AVSA can use the more acoustic information than the pressure sensor array, and thus the coupling magnified BCAVSA can provide lower sidelobes and a narrower mainlobe than those of the bi-cone pressure sensor array. In addition, it can be seen from Figure 11 that the beampattern of the coupling magnified BCAVSA in the elevation plane has lower sidelobes than that of the coupling magnified bi-cone pressure sensor array. In addition, the coupling magnified bi-cone pressure sensor array and the coupling magnified BCAVSA have the same width of the mainlobe. This is because when the elevation angle is equal to 90 degrees, the magnification effect of the coupling system plays a stronger role than the AVSA. In addition, from Figure 10a and Figure 8a, we can see that when the elevation angle is invariant, and the beampatterns in Figure 10a and Figure 8a have the same beamwidth in the azimuth plane. This is because the directivity function of the circular pressure sensor array is one part of the cylindrical AVSA. As the coupling BCAVSA has a narrower mainlobe and lower sidelobes compared to the coupling bi-cone pressure sensor array, the denominator in Equation (Equation 31) associated with coupling BCAVSA is smaller than that of the bi-cone pressure sensor array. Thus, according to Equation (Equation 31), the AG of the coupling BCAVSA is higher than the coupling bi-cone pressure sensor array.

Figure 12 shows the beampatterns of the coupling magnified BCAVSA and bi-cone pressure sensor array in the elevation plane, where the azimuth angle is equal to 130°. The main beam is targeted at $\phi_{s}=80{}^{\circ }$ and $\theta_{s}=130{}^{\circ }$. It can be seen from Figure 12 that the coupling magnified BCAVSA’s beampattern has lower sidelobes and a narrower mainlobe than those of the coupling magnified bi-cone pressure sensor array. This is because the AVSA can use more acoustic information than the pressure sensor array. The advantage of the AVSA plays a stronger role than the magnification effect of the coupling system when the elevation angle is not equal to 90°. In this condition, since the coupling BCAVSA has a narrower mainlobe and lower sidelobes compared to the coupling bi-cone pressure sensor array, the denominator in Equation (Equation 31) associated with coupling BCAVSA is smaller than that of the bi-cone pressure sensor array. As a result, the AG of the coupling BCAVSA is higher than the coupling bi-cone pressure sensor array.

### 4.3. Frequency Invariant Beampattern of BCAVSA over Multiple Octaves

A BCAVSA composed of three cylindrical AVSAs is considered. The desired bandwidth of the BCAVSA is equal to f∈[250 Hz,1000 Hz]. The desired bandwidth can be divided into two subbands, the first and the second cylindrical AVSAs are used to form the constant beamwidth pattern in [500 Hz,1000 Hz], and the second and third cylindrical AVSAs are used to construct the constant beamwidth pattern in [250 Hz,500 Hz]. The radii of the three cylindrical AVSAs are equal to $r_{1}=0.225\;m=0.15\lambda_{H,2}$, $r_{2}=0.45\;m=0.3\lambda_{H,2}$, and $r_{3}=0.9\;m=0.3\lambda_{H,1}$, respectively, $\lambda_{H,1}=c/f_{H,1}\left(f_{H,1}=500\;\mathrm{Hz}\right)$, $\lambda_{H,2}=c/f_{H,2}\left(f_{H,2}=1000\;\mathrm{Hz}\right)$. The two UCAVSAs of the three cylindrical AVSAs are located at $z_{1}=\mp 0.075\;m=0.05\lambda_{H,2}$, $z_{2}=\mp 0.15\;m=0.1\lambda_{H,2}$, and $z_{3}=\mp 0.3\;m)=0.1\lambda_{H,1}$, respectively. The constant beamwidth pattern over this desired bandwidth is constructed by using the three cylindrical AVSAs. The sound speed in the underwater is equal to 1500 m/s. The beam is targeted at $\theta_{s}=180{}^{\circ }$, $\phi_{s}=90{}^{\circ }$. Since the beampattern is a function of three variables *f*, θ, and ϕ, we only provide the beampatterns at some frequency bins. Figure 13 shows the normalized frequency invariant beampatterns of the coupling magnified BCAVSA, which are associated with the frequency 400 Hz and 900 Hz. In addition, Figure 14 shows the frequency invariant beampattern over the frequency [250 Hz,1000 Hz] in the azimuth plane, where the elevation angle is fixed to 90°. Figure 15 shows the frequency invariant beampattern over the frequency [250 Hz,1000 Hz] in the elevation plane, where the azimuth angle is fixed to 180°.

It can be seen from Figure 13a,b that the normalized beampatterns at f=400 Hz and f=900 Hz have approximately the same beamwidth. Form Figure 14 and Figure 15, it can be seen that, by using the three cylindrical AVSAs, the proposed method can form the frequency invariant beampattern over the two octaves [250 Hz,1000 Hz] in the azimuth and elevation planes. The frequency invariant property can be verified by Figure 13, Figure 14 and Figure 15. This simulation case illustrates that, by using multiple cylindrical AVSAs constituting the BCAVSA, the proposed method can form the frequency invariant beampattern over multiple octaves in the azimuth and elevation planes.

### 4.4. Frequency Invariant Beamforming in the Presence of the Noise

The coupling magnified system not only amplifies the phase and amplitude differences of the incident signal but also the ambient noise [35]. In order to reduce the influence of the coupling magnified system to the noise, in this paper, we first construct the covariance matrix $R_{vv,q}$ of the *q*th cylindrical AVSA (q=1,⋯,Q). The covariance matrix $R_{vv,q}$ is decomposed into signal subspace $E_{s}$ and noise subspace $N_{n}$. We then use the signal subspace $E_{s}$ and the eigenvalue corresponding to the signal power to reconstruct a new covariance matrix $R_{new}=E_{s}D_{s}E_{s}^{H}$, from which the influence of noise has theoretically been removed. Subsequently, we introduce the coupling magnified matrix to $R_{new}$ to obtain the covariance matrix of the virtual cylindrical AVSA with larger aperture. Finally, we use the frequency weighted coefficients to obtain the constant beamwidth pattern.

Figure 16 and Figure 17 show the frequency invariant beampattern, where signal-to-noise (SNR) is equal to 5 dB and −5 dB, respectively. The noise received by the BCAVSA is the white Gaussian noise. The other simulation conditions are as same as those in Section 4.1. It can be seen from Figure 16 that, when the SNR is equal to 5 dB, the proposed method can maintain the perfect constant beamwidth beampatterns on the azimuth and elevation planes. Form Figure 17, it can be seen that, when the SNR decreases, the proposed frequency invariant beamforming can form the perfect constant beamwidth pattern in the azimuth plane. In addition, the proposed method can keep a good constant beamwidth beampattern within the mainlobe on the elevation plane, although the width outside the mainlobe of the beampattern at different frequencies are different. From Figure 16 and Figure 17, it illustrates that the proposed frequency invariant beamforming method is not very sensitive to the noise.

Figure 18 shows AG curves of the coupling magnified BCAVSA, the coupling magnified bi-cone pressure sensor array, and the original BCAVSA, where SNR is set to be −5 dB to 10 dB with the interval of 1 dB. The AG curves are the average of 50 times of independent computer simulations. It can be seen from Figure 18 that the coupling magnified BCAVSA has higher AG compared to the coupling magnified bi-cone pressure array and original BCAVSA. This is because, based on advantages of the coupling magnified system and AVSs, the coupling magnified BCAVSA provides the lower sidelobes and a narrower mainlobe than the coupling magnified bi-cone pressure array and original BCAVSA, and thus, according to Equation (Equation 31), the coupling magnified BCAVSA has higher AG than the other methods.

### 4.5. Frequency Invariant Beamforming in the Presence of the Steering Vector Error

This section considers the robustness of the proposed frequency invariant beamforming method in the presence of the steering vector error. In this simulation, the steering vector error for each cylindrical AVSA is defined as:(32)$$\Delta =\chi \left({\overset{ˇ}{a}}_{c}*e^{j\vartheta }\right)⊗\left({\overset{ˇ}{a}}_{l}*e^{j\varphi }\right),$$
where χ is the coefficient to modify the mismatch error. ∗ denotes the Hadamard product. ${\overset{ˇ}{a}}_{c}$ and ${\overset{ˇ}{a}}_{l}$ denote the N×1 and 2×1 Gaussian random process with the mean of zero and variance of 1, respectively. ϑ is the uniformly distributed random process in [0,2π]. φ is the uniformly distributed random process in [0,π].

Figure 19 and Figure 20 show the frequency invariant beampattern in the presence of the steering vector error, where χ is equal to 0.001 and 0.01, respectively.

It can be seen from Figure 19 that, when the amplitude of the steering vector error is equal to 0.001, the proposed frequency invariant beamforming can hold the perfect constant beamwidth on the azimuth and elevation planes. When the amplitude of the steering vector error is equal to 0.01, it can be seen from Figure 20 that the proposed frequency invariant beamforming can hold a good constant beamwidth on the azimuth plane. However, the proposed method cannot hold the constant beamwidth on the elevation plane, and it can be also seen that, for some frequency bins, the mainlobe is not targeted at the 90°. The main reason is that, for the elevation plane, the cylindrical AVSA is equivalent to a two-element array, and it is difficult to achieve robustness using too less number of elements. It illustrates that the proposed method can maintain the perfect constant beamwidth pattern on the azimuth plane in the presence of the steering vector error. In addition, the proposed frequency invariant beampattern on the elevation plane is sensitive to the steering vector error.

## 5. Conclusions

In this paper, we present an approach of designing a small-sized BCAVSA and a theory and method to form a characteristic of the nearly constant beamwidth in the directivity pattern. By appropriately adjusting the coupling parameters, the coupling magnified system of the *Ormia ochracea*’s two ears is extended into the AVSA, and the coupling magnified matrix for the cylindrical AVSAs is derived in order to amplify the phase difference of the two adjacent AVSs. The cylindrical AVSAs with different radii, heights, and the same number of AVSs can be viewed as units which constitute the virtual BCAVSA. By extending the principle of the harmonic nesting to the several groups of cylindrical AVSAs, the constant beamwidth beamforming method is developed. The simulation results demonstrate the following aspects:

(1) Compared to the original BCAVSA, the coupling magnified BCAVSA can provide a directivity pattern with a narrower beamwidth. This is because the coupling magnified system can convert the original BCAVSA into a virtual BCAVSA with larger interelement spacing.

(2) Since the directivity function of the circular array is one part of the cylindrical AVSA, the frequency invariant beampattern has the same beamwidth over the 360 degrees in the azimuth plane when the elevation is fixed.

(3) Since AVS can use more acoustic information than pressure sensors, the coupling magnified BCAVSA has lower sidelobes compared to the coupling magnified bi-cone pressure sensor array in both of the elevation and azimuth planes. In addition, the beampattern in azimuth plane of the coupling magnified BCAVSA has a narrower mainlobe compared to that of the coupling magnified bi-cone pressure sensor array. In the elevation plane, the coupling magnified BCAVSA has a narrower mainlobe compared to that of the coupling magnified bi-cone pressure sensor array when the mainpattern is not targeted at the elevation angle $\phi =90{}^{\circ }$ since the advantage of the AVSA plays a stronger role than the effect of a coupling magnified system; when the main beam is targeted at the elevation angle $\phi =90{}^{\circ }$, the coupling magnified BCAVSA and bi-cone pressure sensor array have the same beamwidth, which is because the magnification effect of the coupling system plays a stronger role than the AVSA.

(4) By using multiple cylindrical AVSAs to constitute the BCAVSA, the proposed method can form the frequency invariant beampattern on the azimuth and elevation planes over multiple octaves.

(5) As the proposed method is based on advantages of the coupling magnified system and AVSs, the coupling magnified BCAVSA has higher AG than the other methods.

(6) In order to reduce the influence of the coupling magnified system to the noise, in this paper, we use the signal subspace and the eigenvalue corresponding the signal subspace to construct a new matrix before applying the coupling magnified system. It is found that the proposed beamforming method can still obtain a perfect constant beamwidth pattern when the SNR is low.

(7) For the small steering vector error, the proposed method can hold a good constant beamwidth pattern. However, when the steering vector error is relatively large, the proposed method cannot maintain a good constant beamwidth pattern on the elevation plane, although it can keep a good constant beamwidth pattern on the azimuth plane.

Finally, in our future work, we will discuss the influence of the electromagnetic coupling for the small sensor arrays and study the frequency invariant beamforming under the condition of the electromagnetic coupling. In addition, the method to improve the robustness of this method will also be studied in the future.

## Figures and Tables

**Figure 1 sensors-20-00661-f001:**
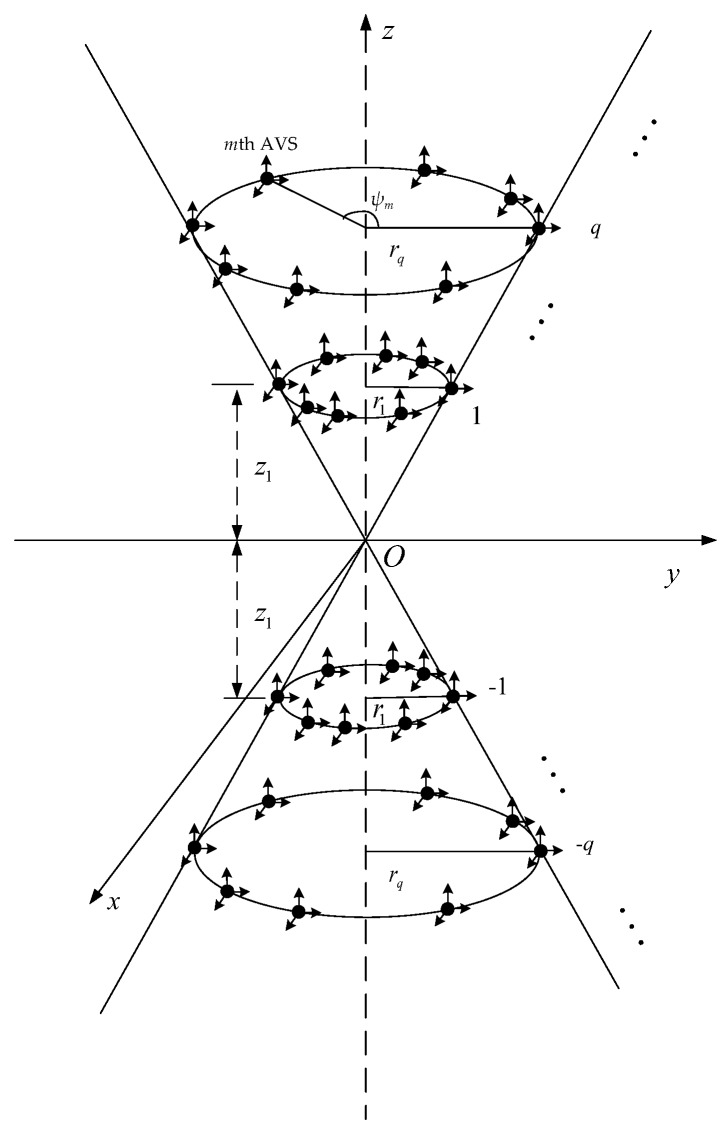
Geometry diagram of the BCAVSA.

**Figure 2 sensors-20-00661-f002:**
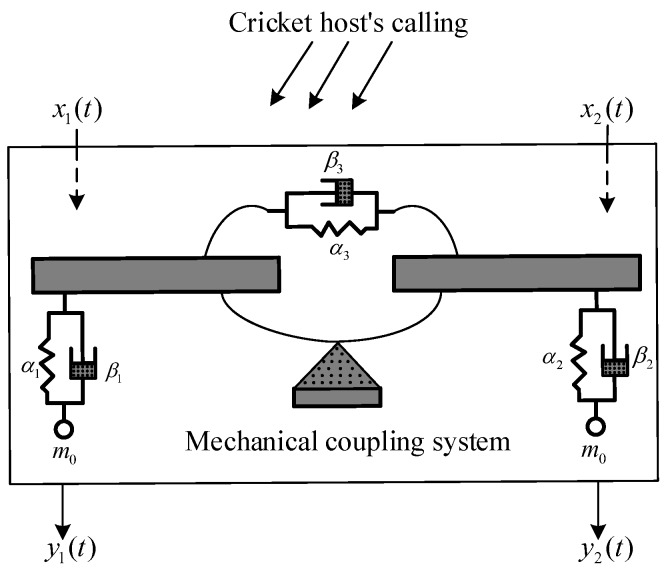
Mechanical model of the *Ormia ochracea*’s two ears [28].

**Figure 3 sensors-20-00661-f003:**
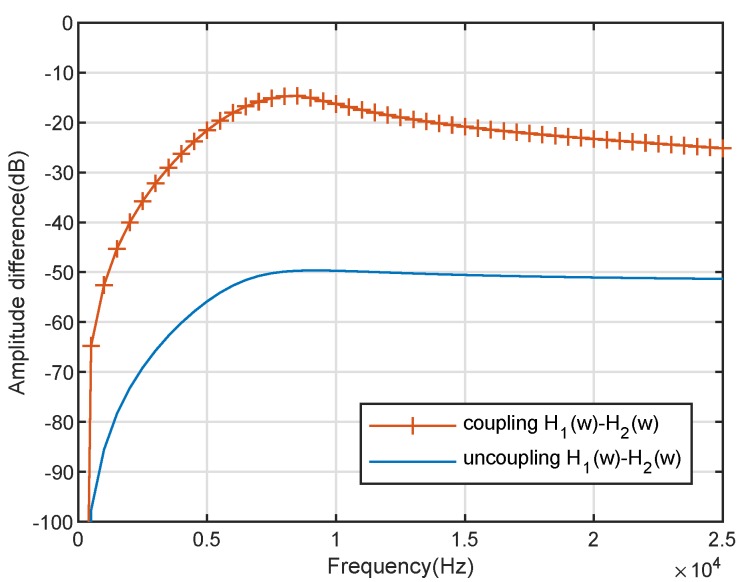
Amplitude difference of the *Ormia ochracea*’s two ears with coupling and uncoupling.

**Figure 4 sensors-20-00661-f004:**
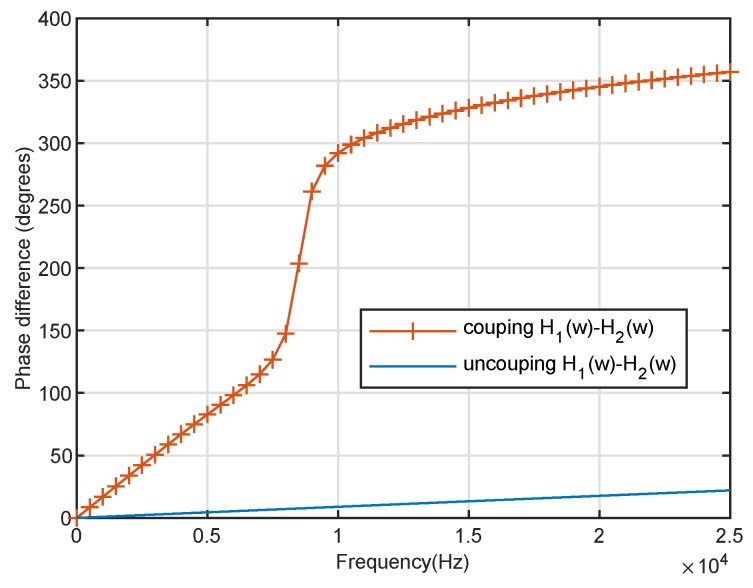
Phase difference of the *Ormia ochracea*’s two ears with coupling and uncoupling.

**Figure 5 sensors-20-00661-f005:**
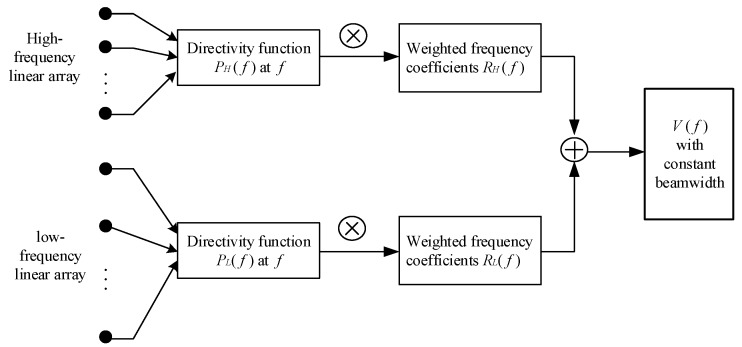
The example of explaining the harmonic nesting method.

**Figure 6 sensors-20-00661-f006:**
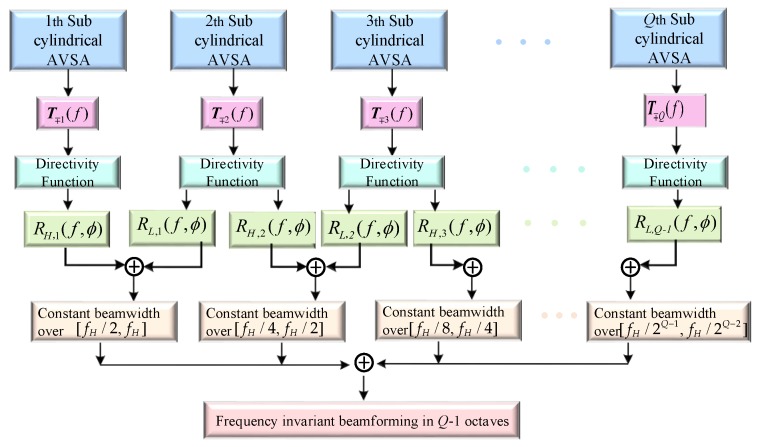
Geometry of the frequency invariant beamforming in multiple octaves.

**Figure 7 sensors-20-00661-f007:**
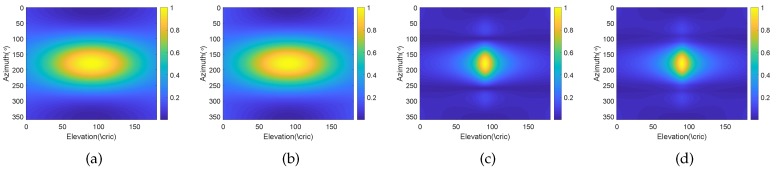
Normalized beampattern at 500 Hz and 700 Hz. (**a**) Original BCAVSA with f=500 Hz; (**b**) original BCAVSA with f=700 Hz; (**c**) coupling magnified BCAVSA with f=500 Hz; and (**d**) coupling magnified BCAVSA with f=700 Hz.

**Figure 8 sensors-20-00661-f008:**
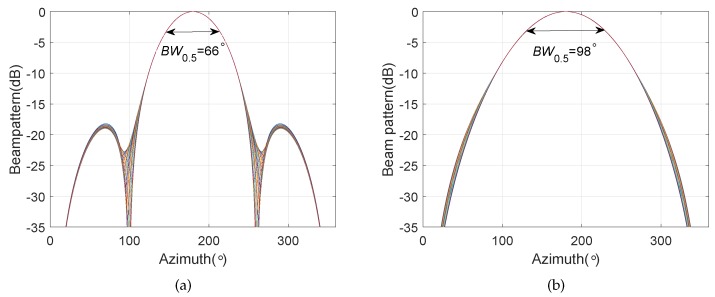
Beampattern in the azimuth plane with the elevation angle $\phi =90{}^{\circ }$, where the beampattern is steered at $\theta_{s}=180{}^{\circ }$. (**a**) Coupling magnified BCAVSA; (**b**) original BCAVSA.

**Figure 9 sensors-20-00661-f009:**
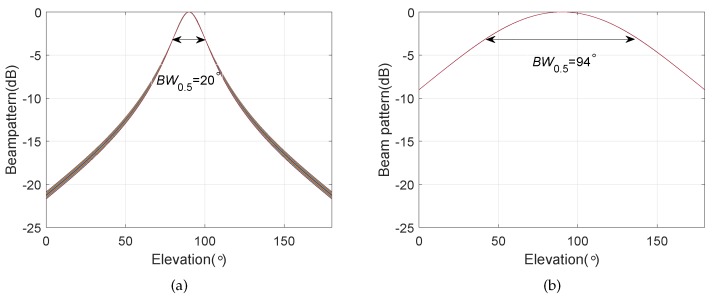
Beampattern in the elevation plane with the azimuth angle $\theta =180{}^{\circ }$, where the beampattern is steered at $\phi_{s}=90{}^{\circ }$. (**a**) Coupling magnified BCAVSA; (**b**) original BCAVSA.

**Figure 10 sensors-20-00661-f010:**
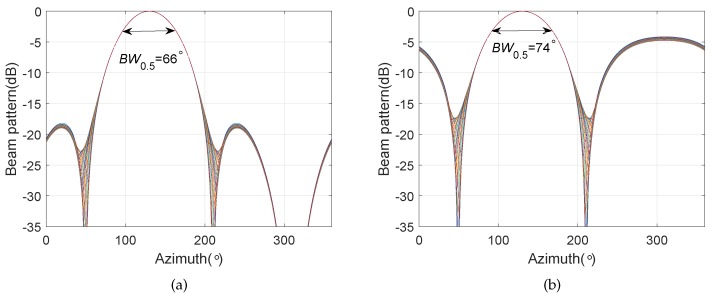
Beampatterns of the coupling magnified array in the azimuth plane with the elevation angle $\phi =90{}^{\circ }$, where the beampattern is steered at $\theta_{s}=130{}^{\circ }$. (**a**) BCAVSA; (**b**) bi-cone pressure sensor array

**Figure 11 sensors-20-00661-f011:**
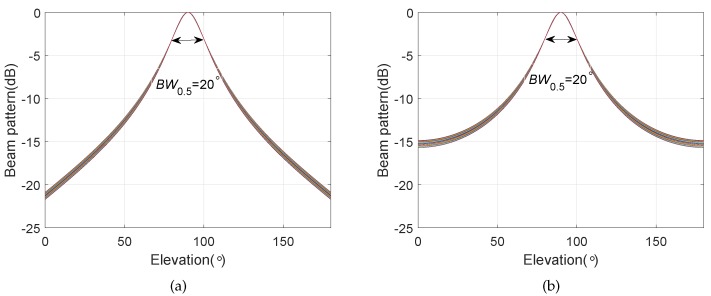
Beampatterns of the coupling magnified array in the elevation plane, where the beampattern is steered at $\phi_{s}=90{}^{\circ }$ and the azimuth angle is fixed at $\theta =130{}^{\circ }$. (**a**) BCAVSA; (**b**) bi-cone pressure sensor array.

**Figure 12 sensors-20-00661-f012:**
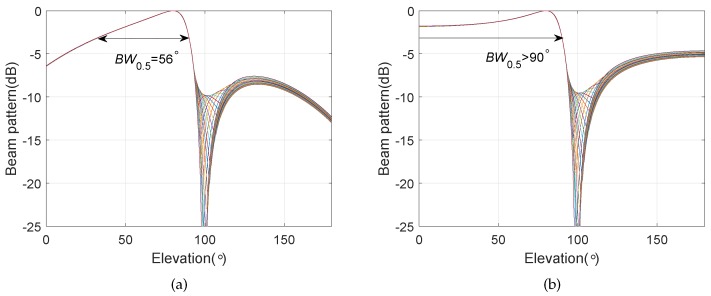
Beampattern in the elevation plane, where the beampattern is steered at $\phi_{s}=80{}^{\circ }$ and the azimuth angle is fixed to 130°. (**a**) BCAVSA; (**b**) bi-cone pressure sensor array.

**Figure 13 sensors-20-00661-f013:**
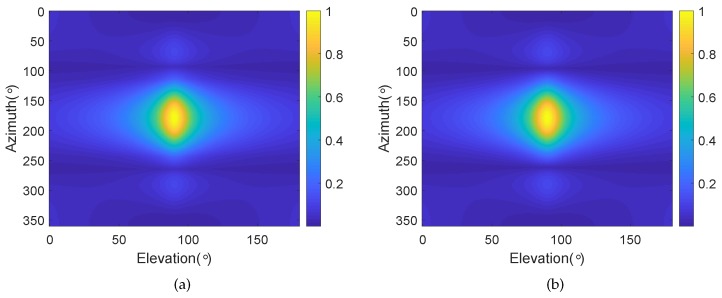
Normalized frequency invariant beampatterns at f=400 Hz and f=900 Hz. (**a**) f=400 Hz; (**b**) f=900 Hz.

**Figure 14 sensors-20-00661-f014:**
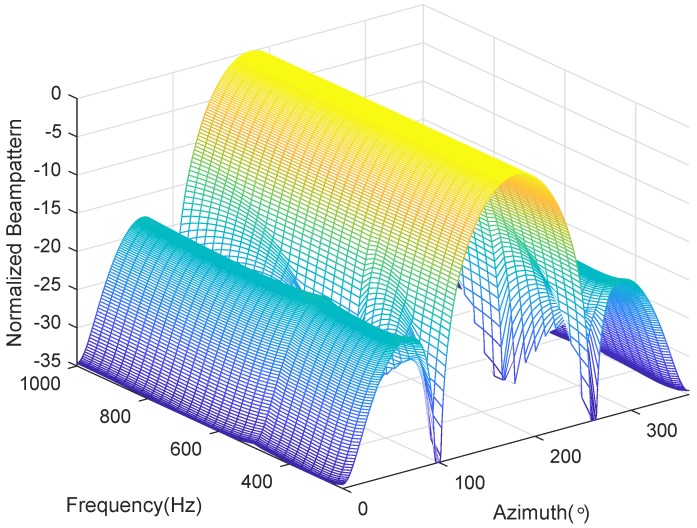
Frequency invariant beampattern over [250 Hz,1000 Hz] in the azimuth plane, where the elevation angle is fixed to 90°.

**Figure 15 sensors-20-00661-f015:**
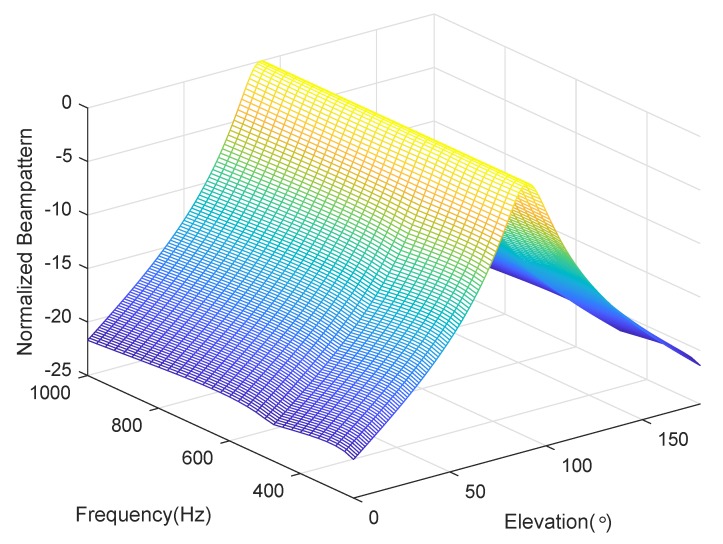
Frequency invariant beampattern over [250 Hz,1000 Hz] in the elevation plane, where the azimuth angle is fixed to 180°.

**Figure 16 sensors-20-00661-f016:**
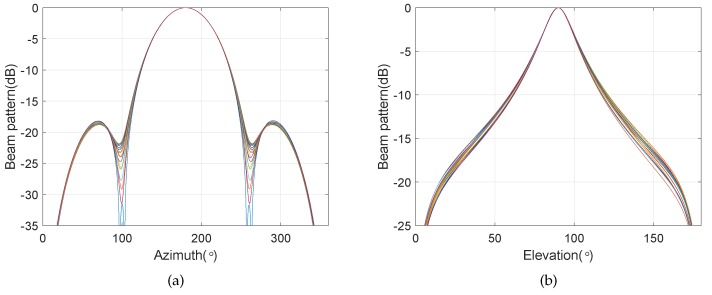
The beampatterns on the azimuth and elevation planes, where SNR is equal to 5 dB. (**a**) Beampattern on the azimuth plane; (**b**) beampattern on the elevation plane.

**Figure 17 sensors-20-00661-f017:**
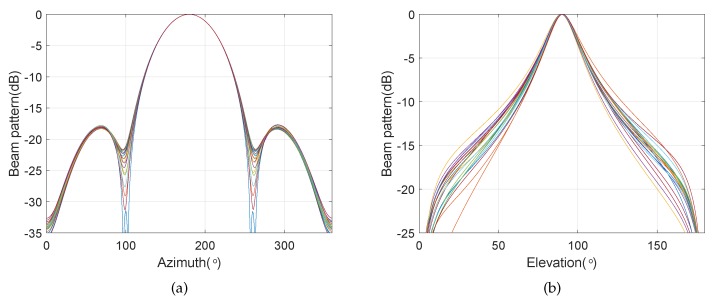
The beampatterns on the azimuth and elevation planes, where SNR is equal to −5 dB. (**a**) Beampattern on the azimuth plane; (**b**) beampattern on the elevation plane.

**Figure 18 sensors-20-00661-f018:**
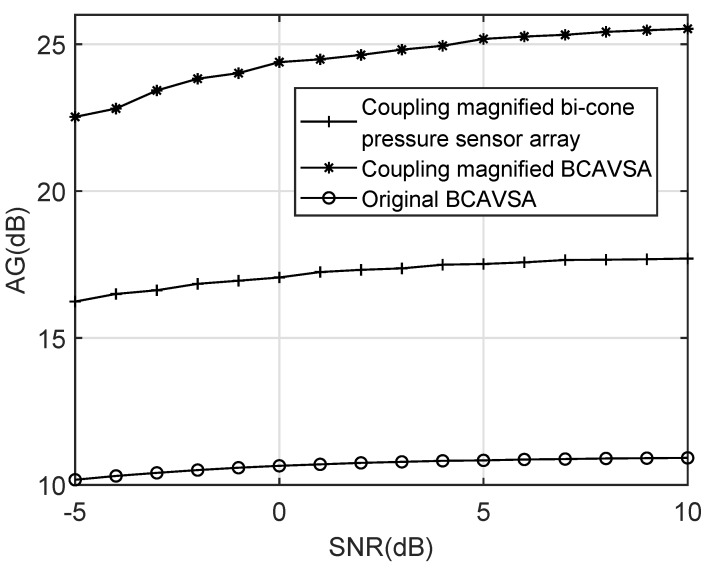
AG curves versus SNR.

**Figure 19 sensors-20-00661-f019:**
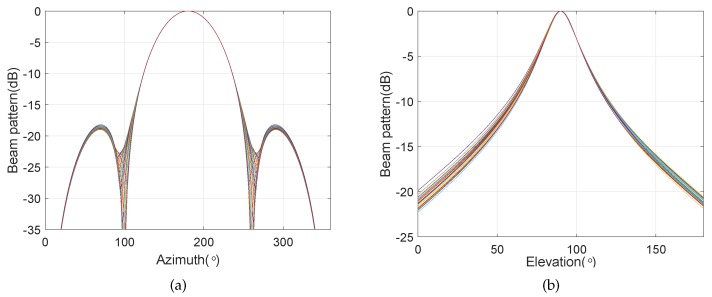
The beampatterns on the azimuth and elevation planes in the presence of steering vector error, where the amplitude of the error is equal to 0.001. (**a**) Beampattern on the azimuth plane; (**b**) beampattern on the elevation plane.

**Figure 20 sensors-20-00661-f020:**
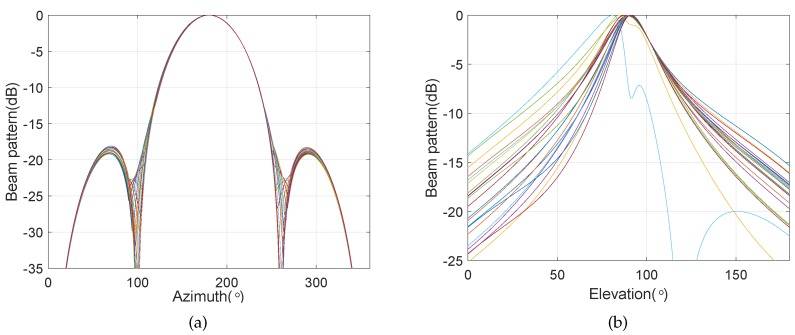
The beampatterns on the azimuth and elevation planes in the presence of steering vector error, where the amplitude of the error is equal to 0.01. (**a**) Beampattern on the azimuth plane; (**b**) beampattern on the elevation plane.

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
