# Peer review of "Frequency Invariant Beamforming for a Small-Sized Bi-Cone Acoustic Vector–Sensor Array"

_sensors, 2020, doi:10.3390/s20030661_

Round 1

Reviewer 1 Report

Dear authors,

First of all, congratulations for your work.

Now, here are my comments to your paper:

Line 72: Do you mean "planar" arrays, instead of "planer" arrays? Section 2.1: You could include phi_m in figure 1.

Line 124: It could be interesting studying the undesired electromagnetic coupling. At least, you can include that as future work in the conclusions section.

Lines 130-131: It would be better to present the definition of D3(f) before the G(f) one. This order is more coherent.

Equation 11: What is the size of these matrices? It is not clear how "the ones" are distributed inside Ac matrix.

Line 165: What are the values of do and fo? The same that are shown on lines 157 and 159? This is not clear.

Section 3.2: You should explain the harmonic nested method, with an easy example, and after that, extrapolate this method to you specific tests. It make this section clearer. Now it is difficult to understand.

The explanations in text of figures 10 and 11 are just the same, but them are really different...  Figure 10 and figure 11 footnotes are also identical... The different is the elevation angle, isn't it? You should explain it better.

Section 4.3: The number of cylindrical AVSAs determines the number of octave bands whose frequency beampattern is invariant, isn't it? What is the specific relation? With Q cylindrical AVSAs you could obtain Q-1 octave bands with constant frequency beampattern? You should explain that. You should also explain if there is any restriction with the frequency values where the beamwidth is constant. The high frequency can be as high as we want?

Conclusions: In points 3 and 4, you talk about AVSAs. It is correct? I mean, do you mean AVSAs or AVSs? It is not clear for me. And the last sentence ("Numerical simutaion...") seems to be part of point 5, and it is related with all the point showed in this section. Although you could elliminate this last sentence...

Author Response

     Thank you very much for the reviewer’ constructive comments concerning our manuscript (ID: sensors-688964). We are very glad that you have given us the opportunity to revise our manuscript. The comments are very valuable and very helpful for revising our manuscript and our further study. According to the reviewer’ comments, we have tried best to modify our manuscript and made the corresponding changes in the revised manuscript, which are highlighted in red. Please see “Respond to reviewer” for the point-to-point responses to the reviewer’s comments.

Reviewer 2 Report

This paper focuses on designing small-sized BCAVSAs with constant beamwidth in the directivity pattern. The coupling magnified system of the Ormia’s two ears is used to amplify the phase difference of the two adjacent AVSs so that the directivity pattern with a narrower beamwidth can be achieved. Several groups of cylindrical AVSAs with different radii are designed using the principle of the harmonic nesting, which can provide directivity patterns with a constant beamwidth. The question is that the authors didn't consider the robustness of their approach. Does the approach still work well in the presence of errors and noise? Is the proposed approach less sensitive to errors than the other methods? Equation (31) seems not be used in the paper. One suggestion is that the authors can add the results of AG or DI of the approach.

Author Response

(The authors gave the same response as above.)

Reviewer 3 Report

Please put a reference under Fig. 2 in the caption. The related work section and the literature review part is thin. It could be expanded. Section 2.1 should be reformatted. It should work as a comprehensive related work to give a background and state-of-the-art to the reader. Appropriate references should be provided. Re Fig. 1: If the design was taken from the literature, it should be clearly mentioned, otherwise, a proof should be provided on why this design is an optimal one for the purpose of this method. Please put appropriate references bellow Eq. 6 for D_1 to D_3. The manuscript emphasized on applying its proposal on a “small-sized BCAVSA”, but there is no discussion/plot on the effect of size on this design and how it changes the beampatterns. Moreover, does it affect the beamformer’s coefficients? Re Eq. 17: Please clarify the parameter of the “max” operations. Moreover, explain how the product of functions in lines 173 and 174 is equivalent with Eq. 17. Check the line numbering and the format. Many of the paragraphs without line numbering and cannot be referred., e.g., lines 172-173, etc. More discussion is required for the figures in the text. In many cases, the authors only have cited the figures without concluding any results that can support the paper's proposal.

Author Response

(The authors gave the same response as above.)
